# Abortion and Female Cancer Risks among Women Aged 20 to 45 Years: A 10-Year Longitudinal Population-Based Cohort Study in Taiwan

**DOI:** 10.3390/ijerph20043682

**Published:** 2023-02-19

**Authors:** Cheng-Ting Shen, Shu-Yu Tai, Yu-Hsiang Tsao, Fang-Ming Chen, Hui-Min Hsieh

**Affiliations:** 1Department of Family Medicine, Kaohsiung Municipal Ta-Tung Hospital, Kaohsiung Medical University, Kaohsiung City 807, Taiwan; 2Department of Family Medicine, School of Medicine, College of Medicine, Kaohsiung Medical University, Kaohsiung City 807, Taiwan; 3Division of Medical Statistics and Bioinformatics, Department of Medical Research, Kaohsiung Medical University Hospital, Kaohsiung Medical University, Kaohsiung City 807, Taiwan; 4Department of Surgery, Kaohsiung Municipal Ta-Tung Hospital, Kaohsiung Medical University, Kaohsiung City 807, Taiwan; 5Department of Surgery, Faculty of Medicine, College of Medicine, Kaohsiung Medical University, Kaohsiung City 807, Taiwan; 6Department of Public Health, Kaohsiung Medical University, Kaohsiung City 807, Taiwan; 7Department of Medical Research, Kaohsiung Medical University Hospital, Kaohsiung City 80756, Taiwan; 8Department of Community Medicine, Kaohsiung Medical University Hospital, Kaohsiung City 80756, Taiwan; 9Center for Big Data Research, Kaohsiung Medical University, Kaohsiung City 807, Taiwan; 10Research Center for Environmental Medicine, Kaohsiung Medical University, Kaohsiung City 807, Taiwan

**Keywords:** female cancer risk, abortion, women’s health

## Abstract

Background: Female cancers, including breast, cervical, uterine, and ovarian cancer, remain among the ten most common cancers among women worldwide, but the relationship between female cancers and abortion from previous studies is inconsistent. This study aimed to investigate risks of incident female cancers among women aged 20 to 45 years who underwent abortion in Taiwan compared with those who did not. Method: A longitudinal observational cohort study was conducted using three nationwide population-based databases in Taiwan, focusing on 20- to 45-year-old women, with 10 years of follow-up. Matched cohorts were identified with propensity score 1-to-3 matching between 269,050 women who underwent abortion and 807,150 who did not. Multivariable Cox proportional hazard modeling was used for analysis after adjusting for covariates including age, average monthly payroll, fertility, diabetes mellitus, polycystic ovarian syndrome, endometrial hyperplasia, endometriosis, hormone-related drugs, and Charlson comorbidity index. Results: We found lower risk of uterine cancer (hazard ratio [HR]: 0.77, 95% CI: 0.70–0.85) and ovarian cancer (HR: 0.81, 95% CI: 0.75–0.88), but no significant difference in risk of breast cancer or cervical cancer, among matched abortion compared with non-abortion cohorts. Regarding subgroup analysis, cervical cancer risk was higher for parous women who underwent abortion, and uterine cancer risk was lower for nulliparous women who underwent abortion compared with non-abortion groups. Conclusions: Abortion was related to lower uterine and ovarian cancer risk but was not associated with risks of incident breast cancer or cervical cancer. Longer follow-up may be necessary to observe risks of female cancers at older ages.

## 1. Introduction

According to global estimates in 2020, breast cancer was the most common incident cancer and the leading cause of cancer mortality in women. In addition, some gynecological cancers, including cervical cancer, uterine cancer, and ovarian cancer, were among the ten most common female cancers threatening women’s health [1]. In Taiwan, the incidence and mortality rates of breast, uterine, and ovarian cancer increased year after year, while incidence and mortality of cervical cancer gradually declined. The age-standardized incidence rates of female breast cancer, cervical cancer, uterine cancer, and ovarian cancer in Taiwan were 80.99, 7.67, 17.00, and 9.86 per million per year in 2019, respectively [2].

Known risk factors are associated with incident breast cancer and other female cancers. For instance, breast cancer is associated with early menarche, late menopause, nulliparity, first birth at older age, and hormone therapy. Cervical cancer is related to human papillomavirus (HPV) infection, multiple sexual partners, smoking, long-term contraceptive use, and multiple full-term pregnancies. Uterine cancer, or endometrial cancer, is associated with obesity, tamoxifen use, and estrogen therapy after menopause. Factors related to incident ovarian cancer include obesity and childbirth after age 35 [3].

Abortion, including spontaneous abortion and induced abortion, thought of as an interruption of the normal hormone cycle during pregnancy, had been investigated in relation to breast cancer incidence in previous studies, but with mixed results [4,5,6]. Spontaneous abortion was considered a miscarriage, often caused by a defect of the fetus or the maternal environment; induced abortion was defined as termination of pregnancy by a medical procedure [7]. A case-control study conducted in New York focusing on women aged younger than 40 years found elevated breast cancer risk after induced abortion (odds ratios [OR] 1.9) and spontaneous abortion (OR 1.5) [8]. However, a population-based cohort study among 1.5 million Danish women using the national registry of induced abortions found that induced abortion was not related to later incident breast cancer [4]. Another prospective cohort study from the Nurses’ Health Study II including 105,716 young women found that neither induced nor spontaneous abortion was related to breast cancer incidence [5]. However, several studies among Asian populations found an association between abortion and breast cancer risk [9,10]. A meta-analysis focusing on Chinese women found the increasing incident breast cancer risk among women who had induced abortion, particularly with increasing numbers of induced abortions [9]. A case-control study conducted in China found elevated breast cancer incidence among post-menopausal women with a history of medical abortion or higher numbers of surgical abortions [10].

Existing studies examining the association between abortion and other types of female cancers also found inconsistent results. For example, a prospective cohort study among 267,400 female textile workers with nearly 10-year follow-up in China found that abortion history in women was not associated with increased cancer risk, but was related to significantly reduced risk of uterine corpus cancer [6]. A nationwide cohort study among 2,311,332 Danish women found that reduced endometrial cancer risk in those with pregnancy, whether terminated with induced abortion or with childbirth [11]. A case-control study among women aged 50 to 74 years in Sweden found reduced ovarian cancer risk in women with incomplete pregnancies [12], and another case-control study conducted in China found a relationship between lower incident ovarian cancer and women having the history of two or more incomplete pregnancies [13]. Dick et al. (2009) conducted two case-control studies among 4500 Australian women and found no significant association between spontaneous or induced abortions and later incident ovarian cancer for parous or nulliparous women [14]. However, the inconsistent results regarding incident female cancers from different studies may be related to the size of eligible population, study methods, follow-up time, adjusting covariates, and even racial diversity.

Abortion in Taiwan was legalized by the Genetic Health Act of 1985. A woman can undergo abortion only under certain circumstances (e.g., medical reasons, mental health issues, or psychological impact) and must obtain the consent of her husband, or the permission of her parents if she is unmarried and aged younger than 20 years [15]. Although previous studies already examined the relationship between abortion and female cancers, there were fewer recent studies investigating related issues with longer follow-up period, particularly in Asian populations. In addition, there remains a lack of evidence regarding the association between abortion and later potential cancer incidence in Taiwan. This study aimed to investigate the potential risk of abortion regarding further female cancer incidence among fertile women. Specifically, we used a longitudinal population-based cohort study with 10-year follow-up in Taiwan to compare the incident female cancers (i.e., breast, cervical, uterine, and ovarian cancers) between women who did and did not undergo abortion.

## 2. Methods

We conducted a longitudinal observational cohort study using three nationwide population-based databases in Taiwan. The first was the National Health Insurance Research Database (NHIRD), which enrolls more than 99% of Taiwan’s population and contains birth year, sex, monthly payroll, and comorbid conditions, including disease diagnoses and outpatient and inpatient care. The second database was the Taiwan Cancer Registry (TCR), containing records of all types of cancer diagnoses and dates, tracked from 1979 to 2017. The third was the National Death Registry tracked from 1971 to 2017, containing accurate death causes and dates for all populations in Taiwan. We analyzed all data during 2021–2022 in the Health and Welfare Data Science Center of the Ministry of Health and Welfare, a Taiwanese government-operated national data warehouse.

### 2.1. Ethical Aspects

The study followed the ethical standards of the Institutional Review Board of the Kaohsiung Medical University Hospital (IRB number: KMUHIRB-E(I)-20190177) and the Helsinki Declaration of the World Medical Association. Given that these three population-based datasets were all encrypted and de-identified when analyzed under the patient privacy protection regulation of the Health and Welfare Data Science Center of the Ministry of Health and Welfare in Taiwan, patients’ informed consent was waived.

### 2.2. Study Population

We first identified the exposure cohort of women of child-bearing age, aged 20 to 45 years, who had abortion records with an induced or spontaneous abortion diagnosis based on ICD-9-CM diagnosis codes 634 to 637 in an outpatient or inpatient record from the NHIRD between 1 January 2004 and 31 December 2007 (*n* = 278,850). The first date of the abortion record was defined as the index date. For the non-abortion comparison cohort, we included all women aged 20 to 45 years from the NHIRD in 2004 to 2007, then excluded those who had abortion records during the entire study period until the study end date (31 December 2007) (*n* = 5,001,653), in order to clearly investigate the relationship between exposure of abortion and subsequent female cancer events from the comparison of 2 cohorts for at least a 10-year follow-up period. Given that the non-abortion cohort lacked specific event dates, we randomly assigned index dates based on the dynamic frequency distribution of the first date of cohort identification date to the abortion date from the abortion cohort. For both cohorts, women with any cancer diagnosis or death record prior to the index date, or with any values (e.g., birthday or sex) missing from the databases, were excluded. Finally, the abortion cohort (*n* = 269,050) and non-abortion cohort (*n* = 4,715,170) were included for analysis.

To compare the potential female cancer risk between comparable abortion and non-abortion cohorts, the propensity score caliper matching method with 1-to-3 match was used to generate adequate comparison groups based on propensity score in order to reduce the confounding bias from basic characteristics. Propensity score was generated using a logistic regression model including baseline age categories, average monthly payroll groups, fertility, diabetes mellitus, polycystic ovarian syndrome, endometrial hyperplasia, endometriosis, hormone-related drugs, and Charlson comorbidity index (CCI) categories [16,17]. In addition, standardized differences were calculated in covariates between matched cohorts, and all differences less than 10% indicated acceptable matching [16,17]. The final matched cohort included 269,050 women in the abortion group and 807,150 in the non-abortion group. Figure 1 presents the flow chart of inclusion and exclusion criteria in the study population.

### 2.3. Variable Definitions

The major outcome of interest was risks of incident female cancers, including breast, cervical, uterine, and ovarian cancers, comparing matched abortion and non-abortion cohorts. The incident cancer event, which was diagnosed from either seeking medical advice due to physical symptoms or routine screening, was identified as the first date of the female cancer diagnosis from the TCR after the index date and included all cancer stages such as carcinoma in situ and stage I to stage IV. Breast cancer was identified with ICD-9-CM diagnosis code 174 or ICD-10-CM diagnosis code C50, cervical cancer with ICD-9-CM diagnosis code 180 or ICD-10-CM diagnosis code C53, uterine cancer with ICD-9-CM diagnosis code 182 or ICD-10-CM diagnosis code C54, and ovarian cancer with ICD-9-CM diagnosis code 183 or ICD-10-CM diagnosis code C56, C570-C574 to include malignant neoplasm of ovary and other uterine adnexa. To compare groups, we followed each abortion and non-abortion subject for at least 10 years from the index date to the date of incident cancer diagnosis, study end date on 31 December 2007, or death date, whichever came first. We then calculated total person-years for each study subject and cancer incidence rate per 100,000 person-years for each incident cancer event.

Baseline characteristics included in this study were index age in years and age categories (20–24, 25–29, 30–34, 35–39, 40 years or older), average monthly payroll group (dependent, less than NTD 20,000, 20,000–40,000, 40,001 or more), fertility, comorbid diseases (diabetes mellitus, polycystic ovarian syndrome, endometrial hyperplasia, endometriosis), hormone-related drugs, and CCI categories (0, 1, 2 or more). Fertility was identified from the index date to the study end date, with the record of normal spontaneous delivery or cesarean section from the NHIRD. Baseline comorbid diseases, hormone-related drugs (e.g., estrogen, progesterone, or their combination) were derived from the NHIRD one year before and after the index date. Detailed diagnosis codes and medication codes are listed in the Appendix A Table A1.

### 2.4. Statistical Analysis

Student’s *t*-test and a chi-square test were used to compare the means and the proportions of demographic and outcome characteristics between abortion and non-abortion female subjects. Incidence rate ratios were calculated to compare incidence rates per total person-years between the two groups [18]. Differences in incident female cancers between the matched cohorts were analyzed using multivariable Cox proportional hazards models, which were commonly used to measure the relationship between survival time and predictor variables, adjusted for baseline confounding variables. Adjusted hazard ratios (aHR) and 95% confidence intervals (CIs) were reported. To further investigate potential risks of abortion regarding female cancers, we then conducted stratification analysis by each subgroup of demographic or clinical characteristics. All statistical operations were performed using SAS version 9.4; *p* values < 0.05 were considered statistically significant.

## 3. Results

Table 1 reports baseline characteristics among women who did and did not undergo abortion in pre- and post-matching cohorts. Significant differences were found in the baseline characteristics between the pre-matched groups (*p* < 0.001). After propensity score 1-to-3 matching, the two matched groups became comparable. In the abortion cohort, mean age was 28.73 years, 54.25% had given birth, 2.95% had diabetes, 3.50% had polycystic ovarian syndrome, 1.63% had endometrial hyperplasia, 5.40% had endometriosis, and 72.07% had been prescribed hormone-related medications.

Table 2 compares total person-years, crude cancer events, incidence rates per 100,000 person-years, and incidence rate ratios (IRR) for related female cancers between the matched abortion and non-abortion cohorts. The IRRs of uterine and ovarian cancer between matched abortion and non-abortion groups were 0.78 (95% CI: 0.71–0.86) and 0.81 (95% CI: 0.75–0.88), but there was no significant difference in IRR of breast cancer or cervical cancer.

Table 3 presents the full Cox proportional hazard models for investigating the association of abortion and female cancers risk between the matched abortion and non-abortion cohorts after adjusting for covariates. The aHRs of uterine and ovarian cancer were 0.77 (95% CI: 0.70–0.85) and 0.81 (95% CI: 0.75–0.88), and there were no significant effects on risks of breast cancer or cervical cancers. Regarding age categories, the elder women had higher risk across female cancer types when compared with the youngest group. With respect to the fertility factors, parous women had lower risk of cervical cancer, uterine cancer, and ovarian cancer when compared with nulliparous women. Table 4 further shows the stratification results of the effect of abortion on risks of female cancers based on each demographic or clinical characteristic subgroup. Regarding the subgroup of fertility, cervical cancer risk was higher in the abortion cohort compared with non-abortion cohort among the parous group (HR: 1.20, 95% CI: 1.05–1.37) but there was no significant difference among the nulliparous group. The HR of uterine cancer was 0.67 between matched abortion and non-abortion cohorts among the nulliparous group (95% CI: 0.60–0.75) but there was no significant difference among the parous group.

## 4. Discussion

This study included fertile women aged 20 to 45 years between 2004 and 2007, followed up for at least 10 years to investigate the effect of abortion on potential risk of female cancers (breast, cervical, uterine, and ovarian cancers) in Taiwan. The overall results indicate no significant abortion effect on breast or cervical cancer incidence and lower risks of incident uterine and ovarian cancers.

With respect to breast cancer risk, existing studies found mixed evidence related to the associations between abortion and female breast cancers [4,5,19]. It was thought the possible mechanism might be that abortion interrupted the complete differentiation of breast epithelial cells, which was originally promoted by increasing estrogen and progesterone levels during the full-term pregnancy, thus raising the risk of carcinogenesis [10,20]. However, our current study did not find a significant association between abortion and female breast cancer, which is consistent with previous research focusing on young women [5,21]. In addition, similarly to findings from a previous systematic review by Beral et al. (2004) using 53 epidemiological studies in a collaborative group studying hormonal factors in breast cancer, as well as a recent meta-analysis by Tong et al. (2020) [19,22], consistent findings indicated no association between abortion and risk of breast cancer.

Uterine cancer is the most rapidly increasing malignancy and the second most common gynecologic malignancy in Taiwan; 92% of uterine cancer is endometrial cancer, as confirmed by tissue proof according to the cancer registry annual report in Taiwan (2019) [2]. Our findings suggest that abortion may reduce the risk of uterine cancer, consistent with previous studies [6,11,23]. Xu et al. (2004) conducted a population-based case-control study among Shanghai women aged 30 to 69 years and found that women with a record of incomplete pregnancy were associated with reduced endometrial cancer risk (OR: 0.67, 95% CI: 0.53–0.84) [23]. A case-control study found that the incidence of endometrial cancer decreased among women with a history of a number of induced abortions, and also with increasing number of births among Danish women aged 25–49 years [24]. Jordan et al. (2021) conducted a pool analysis study from the Epidemiology of Endometrial Cancer Consortium, which included 11 cohort and 19 case-control studies and found that endometrial cancer risk was related, with 41% reduction with a full-term pregnancy and 7–9% decline with incomplete pregnancy [25]. The protective effect of pregnancy was thought to be related with an early gestational effect with rapidly increasing progesterone levels and high progestogen/estrogen ratio in the first weeks after conception [11]. Although the serum progesterone was up to the level inhibiting mitoses in the first trimester, it was observed to be lower in women who miscarried compared with normal pregnancy [25,26]. The protective effect might be weaker after abortion compared with complete pregnancy. Our subgroup analysis found no significant relationship between uterine cancer and abortion among parous women, but uterine cancer risk was 33% lower in nulliparous women who underwent abortion than among those who did not, which might imply that the protective effect of abortion is independent of a full-term pregnancy.

Our study found that women who underwent abortion may have lower ovarian cancer risk. Existing studies found mixed results [12,13,14,27,28]. For example, a recent study by Lee et al. (2021) observed a pooled analysis including 15 case-control studies conducted in several countries and found that incomplete pregnancies had a protective effect on invasive ovarian cancers (OR = 0.84, 95% CI: 0.79–0.89) [27]. A nationwide case-control study conducted in Denmark found that pregnancy loss was not associated with further ovarian cancer or overall cancer [28]. Another case-control study in the United States also found that incomplete pregnancy had no association with incident ovarian cancer among either nulliparous or parous women, but that a spontaneous abortion before first childbirth was associated with reduced risk (aOR 0.47, 95% CI: 0.30–0.75) [29]. Conversely, an observational study of 274,442 female participants in the European Prospective Investigation into Cancer and Nutrition found higher ovarian cancer risk among women with four or more incomplete pregnancies, but there were relatively few cases in the highest exposure groups [30]. Of several hypotheses regarding ovarian cancer occurrence, the most discussed was the incessant ovulation hypothesis, which proposed that ovulation with repeated rupture and repair of ovulating follicles increased the chance of genetic mutations and potential malignant changes [31]. Nevertheless, the association between abortion and ovarian cancer, and the underlying mechanism thereof, was still not clear [28,32].

In addition, our study did not find significant differences regarding incident cervical cancer between the abortion and non-abortion groups, but did find that higher cervical cancer risk was associated with abortion among parous women in the subgroup analysis. Regarding fertility, parous women had lower risk of uterine cancer and ovarian cancer in our study, which was consistent with previous research on the protective effect of childbirth in uterine cancer and ovarian cancer [26,32]. However, the well-known risk factors for cervical cancer were HPV infection, hormonal contraceptives, and high parity [33]. We found parous women were less likely to have cervical cancer risk than nulliparous women, and further studies may be necessary to investigate the potential confounding factors. The strength of the current study is that it is a nationwide population-based study in Taiwan examining the association between abortion and female cancers. In addition, it is a prospective longitudinal cohort study using three nationwide population-based databases from 2004 to 2007 with 10-year follow-up until 2017, using propensity score matching 1-to-3 for comparison from those without any abortion records during the study period. The records of abortion, deliveries, and comorbid conditions could be derived from the NHIRD, reducing the possibility of selection bias and recall bias. However, our study has some limitations. First, there were several unobservable potential confounding factors due to secondary data analysis, including: family history; lifestyle factors, including cigarette smoking and alcohol; vaccination status, including HPV vaccination; and some hormone-related risk factors, such as age at first childbirth, menarche, and menopause. However, we adjusted for baseline comorbid conditions that may be associated with female cancer occurrence, including diabetes, polycystic ovary syndrome, endometrial hyperplasia, and endometriosis. Although data on hormone-related drugs were collected before and after the index date, we could not completely adjust for their effect due to limited data regarding the dose effect and treatment duration. Second, induced abortion was legal in Taiwan, but some women still may not have sought hospital care after early pregnancy loss, possibly leading to underreporting and underestimating the effect of abortion. In addition, we were not able to distinguish induced abortion from spontaneous abortion in the NHIRD administrative claims. Third, our study population included fertile women, and our findings may not generalize to postmenopausal women. Finally, although the findings in our study may not generalize to other countries, the relationship between abortion and female cancers continues to be investigated in studies from different countries, which implies that this issue remains important for women’s health and is worthy of attention.

## 5. Conclusions

Analyzing the population-based data with a 10-year follow-up period, we found lower uterine and ovarian cancer risk among the matched abortion cohorts compared with non-abortion cohorts, but there was no significant difference in breast cancer or cervical cancer risk. Regarding subgroup analysis, cervical cancer risk was higher among the parous group in matched abortion cohorts, but uterine cancer risk was lower among nulliparous groups. Longer follow-up may be necessary to observe the risks of female cancers among these fertile women at older ages.

## Figures and Tables

**Figure 1 ijerph-20-03682-f001:**
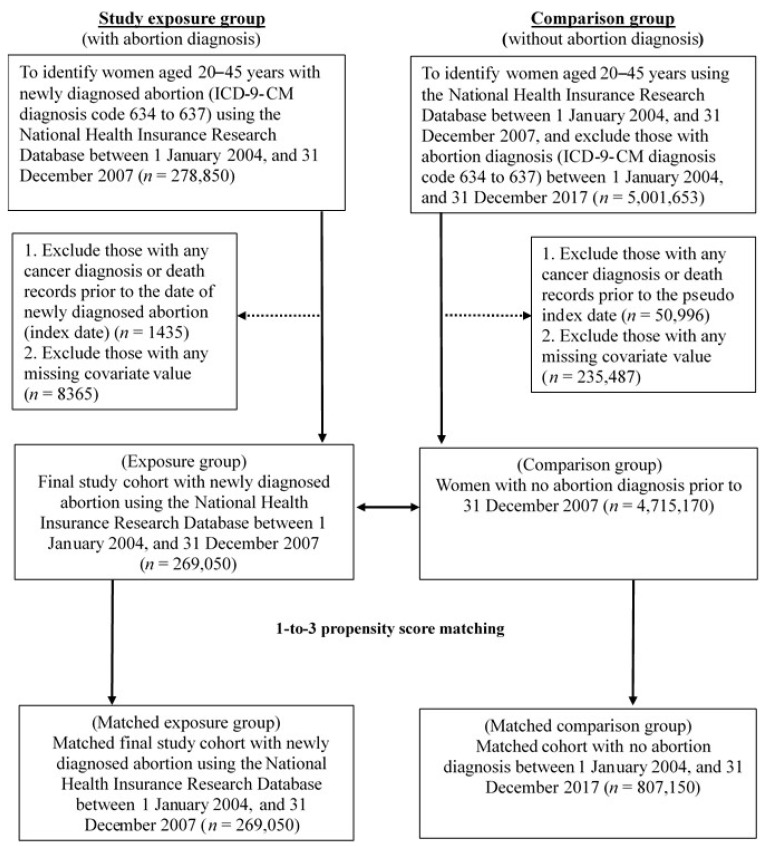
Flow chart of inclusion and exclusion criteria of the study population.

**Table 1 ijerph-20-03682-t001:** Demographic characteristics among abortion and non-abortion cohorts before and after propensity score matching.

	Before PSM Matching		*p*-Value	After PSM Matching		*p*-Value	Standardized Difference
	Abortion	Non-Abortion		Abortion	Non-Abortion		
N	269,050	4,715,170		269,050	807,150		
Age (in years) (Mean±SD)	28.73 (±6.28)	31.45 (±8.02)	<0.001	28.73 (±6.28)	28.75 (±6.46)	0.128	0.3%
Age categories (N, %)^#^			<0.001			1.000	
20–24	81,246 (30.20%)	1,238,399 (26.26%)		81,246 (30.20%)	243,740 (30.20%)		0.0%
25–29	73,636 (27.37%)	848,943 (18.00%)		73,636 (27.37%)	220,938 (27.37%)		0.0%
30–34	59,609 (22.16%)	783,941 (16.63%)		59,609 (22.16%)	178,819 (22.15%)		0.0%
35–39	38,481 (14.30%)	824,727 (17.49%)		38,481 (14.3%)	115,446 (14.3%)		0.0%
40+	16,078 (5.98%)	1,019,160 (21.61%)		16,078 (5.98%)	48,207 (5.97%)		0.0%
Average monthly payroll (NTD, Mean ± SD)	30,621 (±22905)	32,951 (±23776)	<0.001	30,621 (±22905)	30,925 (±23022)	<0.001	1.3%
Average monthly payroll group (N, %) ^#^			<0.001			<0.001	
Dependent	29,818 (11.08%)	509,710 (10.81%)		29,818 (11.08%)	87,691 (10.86%)		0.7%
Less than NTD 20,000	31,390 (11.67%)	536,115 (11.37%)		31,390 (11.67%)	92,337 (11.44%)		0.7%
NTD 20,000–NTD 40,000	153,828 (57.17%)	2,698,492 (57.23%)		153,828 (57.17%)	463,190 (57.39%)		0.4%
NTD 40,001+	54,014 (20.08%)	970,853 (20.59%)		54,014 (20.08%)	163,932 (20.31%)		0.6%
Fertility (N, %) ^#^			<0.001			0.980	
Nulliparous	123,082 (45.75%)	3,404,682 (72.21%)		123,082 (45.75%)	369,269 (45.75%)		0.0%
Parous	145,968 (54.25%)	1,310,488 (27.79%)		145,968 (54.25%)	437,881 (54.25%)		0.0%
Diabetes mellitus (N, %) ^#^			<0.001			0.961	
No	261,105 (97.05%)	4,548,294 (96.46%)		261,105 (97.05%)	783,330 (97.05%)		0.0%
Yes	7945 (2.95%)	166,876 (3.54%)		7945 (2.95%)	23,820 (2.95%)		0.0%
Polycystic ovarian syndrome (N, %) ^#^			<0.001			0.849	
No	259,636 (96.50%)	4,610,982 (97.79%)		259,636 (96.50%)	778,971 (96.51%)		0.1%
Yes	9414 (3.5%)	104,188 (2.21%)		9414 (3.50%)	28,179 (3.49%)		0.1%
Endometrial hyperplasia (N, %) ^#^			<0.001			0.854	
No	264,657 (98.37%)	4,653,461 (98.69%)		264,657 (98.37%)	794,013 (98.37%)		0.0%
Yes	4393 (1.63%)	61,709 (1.31%)		4393 (1.63%)	13,137 (1.63%)		0.0%
Endometriosis (N, %)^#^			<0.001			0.975	
No	254,523 (94.60%)	4,487,591 (95.17%)		254,523 (94.60%)	763,582 (94.60%)		0.0%
Yes	14,527 (5.40%)	227,579 (4.83%)		14,527 (5.40%)	43,568 (5.40%)		0.0%
Hormone-related drugs (N, %) ^#^			<0.001			0.996	
No	75,149 (27.93%)	2,880,486 (61.09%)		75,149 (27.93%)	225,451 (27.93%)		0.0%
Yes	193,901 (72.07%)	1,834,684 (38.91%)		193,901 (72.07%)	581,699 (72.07%)		0.0%
Charlson comorbidity index (Mean ± SD)	0.51 (±0.86)	0.50 (±0.89)	0.001	0.51 (±0.86)	0.51 (±0.86)	0.601	0.0%
Charlson comorbidity index categories (N, %) ^#^			<0.001			0.998	
0 score	174,904 (65.01%)	3,148,004 (66.76%)		174,904 (65.01%)	524,763 (65.02%)		0.0%
1 score	64,880 (24.11%)	1,043,789 (22.14%)		64,880 (24.11%)	194,613 (24.11%)		0.0%
≧2 scores	29,266 (10.88%)	523,377 (11.10%)		29,266 (10.88%)	87,774 (10.87%)		0.0%

Note: NTD = New Taiwan Dollar; PSM = propensity score matching. ^#^ These variables were used in the propensity score matching approach to generate comparable abortion and non-abortion matched cohorts.

**Table 2 ijerph-20-03682-t002:** Comparisons of cancer incidence rates, incidence rate ratios, and hazard ratios for female cancers between matched abortion and non-abortion cohorts.

	Total Person-Years		Cancer Events N (%)			Incidence Rate (per 100,000 Person-Years)		Incidence Rate Ratio		
	Abortion	Non-Abortion	Abortion	Non-Abortion	*p*-Value	Abortion	Non-Abortion	IRR	95% CI	*p*-Value
Breast cancer										
Yes	22,790.78	66,951.09	3006(1.12%)	8869(1.10%)	0.790	92.00	90.45	1.02	0.98–1.06	0.414
No	3,244,557.51	9,737,942.55	266,044(98.88%)	98,281(98.90%)						
Cervical cancer										
Yes	3898.80	10,657.30	654(0.24%)	1798(0.22%)	0.056	19.95	18.28	1.09	1.00–1.19	0.055
No	3,273,542.84	9,825,290.13	268,396(99.76%)	805,352(99.78%)						
Uterine cancer										
Yes	3876.90	15,939.70	564(0.21%)	2171(0.27%)	<0.001	17.20	22.07	0.78	0.71–0.86	<0.001
No	3,275,442.46	9,822,676.97	68,486(99.79%)	804,979(99.73%)						
Ovarian cancer										
Yes	4767.54	17,966.91	802(0.30%)	2966(0.37%)	<0.001	24.48	30.18	0.81	0.75–0.88	<0.001
No	3,270,992.30	9,808,883.83	268,248(99.70%)	804,184(99.63%)						

Note: IRR, incident rate ratio. Covariates listed in the Table 1 were controlled for in the Cox proportional hazard models.

**Table 3 ijerph-20-03682-t003:** Cox proportional hazard full models for female cancer outcomes between matched abortion and non-abortion cohorts.

	Breast Cancer			Cervical Cancer			Uterine Cancer			Ovarian Cancer		
	aHR	95% CI	*p*-Value	aHR	95% CI	*p*-Value	aHR	95% CI	*p*-Value	aHR	95% CI	*p*-Value
Abortion												
Non (Ref)												
Yes	1.01	0.97–1.05	0.719	1.09	0.99–1.19	0.072	0.77	0.70–0.85	<0.001	0.81	0.75–0.88	<0.001
Age categories												
20–24 (Ref)												
25–29	2.51	2.32–2.70	<0.001	1.14	1.02–1.28	0.026	2.28	1.96–2.65	<0.001	1.15	1.05–1.26	0.002
30–34	4.69	4.36–5.05	<0.001	1.40	1.24–1.58	<0.001	3.48	3.01–4.03	<0.001	1.10	1.00–1.21	0.047
35–39	7.17	6.64–7.74	<0.001	1.55	1.36–1.77	<0.001	4.30	3.69–5.02	<0.001	1.11	1.00–1.24	0.056
40+	7.75	7.11–8.45	<0.001	1.54	1.30–1.83	<0.001	4.89	4.12–5.80	<0.001	0.92	0.79–1.07	0.261
Average monthly payroll group												
Less than NTD 20,000 (Ref)												
Dependent	1.19	1.10–1.29	<0.001	0.71	0.61–0.82	<0.001	1.12	0.96–1.31	0.142	1.12	0.98–1.29	0.107
NTD 20,001–NTD 40,000	1.07	1.01–1.15	0.030	0.72	0.65–0.81	<0.001	0.95	0.84–1.08	0.471	1.14	1.02–1.27	0.020
NTD 40,001+	1.32	1.23–1.42	<0.001	0.45	0.38–0.52	<0.001	1.13	0.98–1.30	0.093	1.16	1.02–1.31	0.020
Fertility												
Nulliparous (Ref)												
Parous	1.02	0.98–1.07	0.272	0.80	0.73–0.88	<0.001	0.63	0.58–0.69	<0.001	0.86	0.80–0.93	<0.001
Diabetes mellitus												
No (Ref)												
Yes	0.63	0.57–0.69	<0.001	0.52	0.42–0.64	<0.001	0.84	0.70–1.00	0.047	0.43	0.36–0.51	<0.001
Polycystic ovarian syndrome												
No (Ref)												
Yes	0.93	0.84–1.03	0.172	0.99	0.80–1.23	0.929	1.83	1.57–2.14	<0.001	1.55	1.37–1.77	<0.001
Endometrial hyperplasia												
No (Ref)												
Yes	1.33	1.20–1.49	<0.001	1.53	1.22–1.92	<0.001	3.95	3.46–4.52	<0.001	1.10	0.91–1.34	0.317
Endometriosis												
No (Ref)												
Yes	1.08	1.01–1.16	0.030	0.83	0.71–0.99	0.034	1.61	1.43–1.80	<0.001	3.67	3.38–3.98	<0.001
Hormone-related drugs												
No (Ref)												
Yes	0.91	0.87–0.94	<0.001	1.34	1.22–1.48	<0.001	1.17	1.07–1.28	<0.001	1.11	1.02–1.20	0.012
Charlson comorbidity index categories												
0 score (Ref)												
1 score	1.10	1.05–1.15	<0.001	0.93	0.83–1.04	0.196	1.08	0.98–1.19	0.122	1.05	0.96–1.15	0.318
≧2 scores	2.67	2.55–2.79	<0.001	5.03	4.59–5.50	<0.001	2.49	2.27–2.74	<0.001	4.91	4.56–5.28	<0.001

Note: HR = hazard ratio. Ref = reference group.

**Table 4 ijerph-20-03682-t004:** Stratification analysis of the risks of abortion for incident female cancers from Cox proportional hazard models.

	Breast Cancer			Cervical Cancer			Uterine Cancer			Ovarian Cancer		
	HR	95% CI	*p*-Value	HR	95% CI	*p*-Value	HR	95% CI	*p*-Value	HR	95% CI	*p*-Value
Age categories												
20–24	1.05	0.90–1.21	0.555	1.17	0.97-1.41	0.095	0.90	0.67-1.21	0.472	0.83	0.71-0.97	0.020
25–29	0.97	0.88–1.07	0.503	1.19	0.99–1.42	0.064	0.73	0.59–0.90	0.003	0.74	0.63–0.86	<0.001
30–34	1.03	0.96–1.12	0.352	1.16	0.97–1.38	0.107	0.82	0.70–0.98	0.025	0.84	0.72–0.99	0.034
35–39	0.98	0.91–1.06	0.633	0.87	0.70–1.08	0.204	0.71	0.59–0.85	<0.001	0.88	0.73–1.05	0.149
40+	1.05	0.95–1.17	0.343	0.98	0.73–1.33	0.909	0.79	0.62–1.00	0.049	0.78	0.58–1.05	0.097
Average monthly payroll group												
Dependent	0.94	0.84–1.06	0.306	0.94	0.72–1.24	0.666	0.73	0.57–0.95	0.019	0.88	0.70–1.10	0.258
Less than NTD 20,000	0.98	0.85–1.12	0.723	1.21	0.98–1.50	0.073	0.66	0.49–0.89	0.006	0.82	0.64–1.04	0.097
NTD 20,001–NTD 40,000	1.06	1.00–1.12	0.052	1.06	0.95–1.20	0.300	0.77	0.68–0.88	<0.001	0.80	0.72–0.89	<0.001
NTD 40,001+	1.00	0.92–1.08	0.923	1.18	0.93–1.51	0.183	0.89	0.74–1.06	0.190	0.81	0.69–0.97	0.018
Fertility												
Nulliparous	1.02	0.97–1.08	0.466	1.01	0.90–1.14	0.855	0.67	0.60–0.75	<0.001	0.76	0.68–0.85	<0.001
Parous	1.01	0.94–1.08	0.820	1.20	1.05–1.37	0.007	1.02	0.88–1.19	0.759	0.87	0.78–0.97	0.013
Diabetes mellitus												
No	1.02	0.97–1.06	0.496	1.11	1.02–1.22	0.023	0.78	0.71–0.86	<0.001	0.81	0.75–0.88	<0.001
Yes	1.08	0.88–1.32	0.469	0.68	0.42–1.13	0.135	0.76	0.51–1.14	0.189	0.91	0.60–1.36	0.630
Polycystic ovarian syndrome												
No	1.02	0.98–1.06	0.406	1.10	1.01–1.21	0.035	0.79	0.72–0.87	<0.001	0.83	0.77–0.90	<0.001
Yes	1.00	0.78–1.27	0.980	0.82	0.49–1.36	0.445	0.64	0.43–0.94	0.024	0.57	0.41–0.79	0.001
Endometrial hyperplasia												
No	1.02	0.98–1.07	0.336	1.09	1.00–1.20	0.054	0.82	0.74–0.90	<0.001	0.81	0.75–0.88	<0.001
Yes	0.91	0.70–1.16	0.437	1.03	0.62–1.72	0.898	0.44	0.30–0.64	<0.001	0.91	0.58–1.43	0.691
Endometriosis												
No	1.01	0.97–1.06	0.586	1.10	1.01–1.21	0.037	0.75	0.67–0.82	<0.001	0.83	0.76–0.91	<0.001
Yes	1.08	0.94–1.25	0.286	0.92	0.63–1.35	0.676	1.03	0.81–1.31	0.794	0.74	0.62–0.88	0.001
Hormone-related drugs												
No	0.99	0.92–1.06	0.753	1.21	1.01–1.46	0.043	0.87	0.73–1.04	0.119	0.79	0.67–0.93	0.004
Yes	1.03	0.98–1.09	0.223	1.06	0.96–1.17	0.265	0.75	0.67–0.84	<0.001	0.82	0.75–0.90	<0.001
Charlson comorbidity index categories												
0 score	1.00	0.94–1.06	0.967	1.10	0.97–1.26	0.143	0.72	0.63–0.83	<0.001	0.79	0.71–0.89	<0.001
1 score	1.05	0.96–1.14	0.320	1.16	0.93–1.45	0.190	0.82	0.67–0.99	0.042	0.97	0.81–1.16	0.731
≧2 scores	1.03	0.95–1.12	0.472	1.05	0.91–1.21	0.510	0.86	0.72–1.02	0.080	0.76	0.67–0.87	<0.001

Note: HR = hazard ratio. Covariates listed in Table 1 were controlled for in the Cox proportional hazard models.

## Data Availability

The data that support the findings of this study are available from the Health and Welfare Data Science Center of the Ministry of Health and Welfare, a Taiwanese government-operated national data warehouse, but restrictions apply to the availability of these data, which were used under regulation for the current study, and so are not publicly available. Data are, however, available from the authors upon reasonable request and with permission of the Health and Welfare Data Science Center of the Ministry of Health and Welfare in Taiwan.

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
