# Peer review of "Abortion and Female Cancer Risks among Women Aged 20 to 45 Years: A 10-Year Longitudinal Population-Based Cohort Study in Taiwan"

_ijerph, 2023, doi:10.3390/ijerph20043682_

Round 1

Reviewer 1 Report

The authors have highlighted some major inconsistencies with the reporting of cancer risk for women who have been pregnant or had abortions; many of these may be due to the location of the study. The authors have sought to clarify the risk factor for several cancers within the Taiwanese population.

The authors make the assumption (lines 223-229) that the increased risk of cervical cancer in parous women who had abortions is potentially due to the higher risk of HPV due to "increased sexual behaviors". It is irresponsible of the authors to make the assumption that women who have had abortions are promiscuous, and the authors did not examine HPV infection rate so this statement should be stricken from the manuscript.

The largest concern is that there are no figures nor tables actually included with the manuscript. It is impossible to fully complete this review without that material.

Minor:

-line 58, misspelling of "cancer"

-space missing between words in line 73

-Lines 111-115: the authors have left in the formatting directions; these should be removed.

-Line 260, incident is misspelled

Author Response

Point#1: The authors make the assumption (lines 223-229) that the increased risk of cervical cancer in parous women who had abortions is potentially due to the higher risk of HPV due to "increased sexual behaviors". It is irresponsible of the authors to make the assumption that women who have had abortions are promiscuous, and the authors did not examine HPV infection rate so this statement should be stricken from the manuscript.

Our response: We appreciate the reviewer’s suggestion. We revised the statements in the discussion section as follows:

“In addition, our study did not find significant difference regarding incident cervical cancer between abortion and non-abortion group, but did find that higher cervical cancer risk was associated with abortion among parous women in the subgroup analysis. Regarding fertility, parous women had lower risk in uterine cancer and ovarian cancer from our study, which was consistent as other previous studies about the protective effect of childbirth in uterine cancer and ovarian cancer[26, 33]. However, the well-known risk factors for cervical cancer were HPV infection, hormonal contraceptives, and high parity[23]. We found parous women were less likely to have cervical cancer risk than nulliparous women, and further studies may be necessarily to investigate the potential confounding factors.”

Point#2: The largest concern is that there are no figures nor tables actually included with the manuscript. It is impossible to fully complete this review without that material.

Our response: We appreciate the reviewer’s suggestion. Our tables and figures did include in the original submission, but were not sure why were lost during the journal formation. In the revised edition, we re-attached the tables and figures along with the main text.

Point#3: -line 58, misspelling of "cancer"

Our response: We appreciate the reviewer’s suggestion. We corrected the misspelling and as follows:” Factors related to incident ovarian cancer include obesity and childbirth after age 35 years ”

Point#4: -space missing between words in line 73

Our response: We appreciate the reviewer’s suggestion. We corrected the space missing and as follows “ A meta-analysis focusing on Chinese women found the increasing incident breast cancer risk among women had induced abortion, particularly with increasing numbers of induced abortions.”

Point#5: -Lines 111-115: the authors have left in the formatting directions; these should be removed.

Our response: We appreciate the reviewer’s suggestion. We removed the formatting directions as in the ijerph-2193634 manuscript of line 111-115.

Point#6: -Line 260, incident is misspelled

Our response: We appreciate the reviewer’s suggestion. We corrected the misspelled word and as follows: “ Another case-control study in the United States also found that incomplete pregnancy had no association with incident ovarian cancer among either nulliparous or parous women”.

Reviewer 2 Report

Dear Authors,

The MS entitled " Abortion and Further Female Cancer Risks among Women Aged 20 to 45 Years: a 10-Year Longitudinal Population-Based Cohort Study in Taiwan" was thoroughly reviewed. The MS describes a co-relation between risk of various types of cancers in female (20 to 45 years age) and abortion in Taiwan. The MS is a short study with apparently, dealing only in data. It lacks the basic experimental evidence however, could be appropriate in case studies or short communication.

However, in current form it has many flaws. A major revision is needed. My comments are:

1. Remove lines 111 to 115.

2. Also, no tables or figures are present in the MS which makes the data analysis difficult.

3. the references, citations and formatting should be revised throughout.

4. Introduction is sufficient.

5. The results however are insufficient. They need to be elaborated/expended and more critical analysis with comparison to available literature should be done.  

Author Response

Point#1: Remove lines 111 to 115.

Our response: We appreciate the reviewer’s suggestion. We moved the formatting directions as in the ijerph-2193634 manuscript of line 111-115.

Point#2: Also, no tables or figures are present in the MS which makes the data analysis difficult.

Our response: We appreciate the reviewer’s suggestion. Our tables and figures did include in the original submission, but were not sure why were lost during the journal formation. In the revised edition, we re-attached the tables and figures along with the main text.

Point#3: the references, citations and formatting should be revised throughout.

Our response: We appreciate the reviewer’s suggestion. We will revise the references, citations and formatting.

Point#4: Introduction is sufficient.

Our response: We appreciate the reviewer’s encouragement.

Point#5: The results however are insufficient. They need to be elaborated/expended and more critical analysis with comparison to available literature should be done.  

Our response: We appreciate the reviewer’s suggestion. We made the adjustment of the Results and Discussion and as follows:

  • In the Result Section

“Table 3 presents the full Cox proportional hazard models for investigating the association of abortion and female cancers risk between the matched abortion and non-abortion cohorts after adjusting for covariates. The aHRs of uterine and ovarian cancer were 0.77 (95% CI: 0.70-0.85) and 0.81 (95% CI: 0.75-0.88), and there were no significant effects on risks of breast cancer or cervical cancers. Regarding age categories, the elder women had higher risk across female cancer types when compared with the youngest group. With respect to the fertility factors, parous women had lower risk of cervical cancer, uterine cancer, and ovarian cancer when comparing with nulliparous women. Table 4 further shows the stratification results of the effect of abortion on risks of female cancers based on each demographic or clinical characteristic subgroup. Regarding the subgroup of fertility, cervical cancer risk was higher in abortion cohort while compared with non-abortion among the parous group (HR: 1.20, 95% CI: 1.05-1.37) but there was no significant difference among the nulliparous group. The HR of uterine cancer was 0.67 between matched abortion and non-abortion cohorts among the nulliparous group (95% CI: 0.60-0.75) but there was no significant difference among the parous group.”

  • In the Discussion Section

“ In addition, our study did not find significant difference regarding incident cervical cancer between abortion and non-abortion group, but did find that higher cervical cancer risk was associated with abortion among parous women in the subgroup analysis. Regarding fertility, parous women had lower risk in uterine cancer and ovarian cancer from our study, which was consistent as other previous studies about the protective effect of childbirth in uterine cancer and ovarian cancer[26, 33]. However, the well-known risk factors for cervical cancer were HPV infection, hormonal contraceptives, and high parity[23]. We found parous women were less likely to have cervical cancer risk than nulliparous women, and further survey may be necessarily to investigate the potential confounding factors.”

Reviewer 3 Report

Dear authors, it is an interesting article, try to find increasing risk of cancer with abortion is still an unexplored field.

Some general remarks

Study population

Line 124  Al of theme was sexually active? All of theme was pregnant almost ones?

Please clarify what incident diagnostic means. It includes cancer detected on in a routine screening?

Pleas clarify that if in the dataset any stage of cancer was included eg. Stage I was excluded or included in situ (HSIL) in cervical cancer. 

In cervical cancer do you know the status of vaccination against HPV of participants?

Results

Table1, 2, 3 and flowchart are not included in the manuscript or as an additional material

Discussion

Could you explain, which mechanism could correlate abortion and breast cancer? Long exposure of breast tissue to oestrogens?

Author Response

Point#1 Study population: Line 124  Al of theme was sexually active? All of theme was pregnant almost ones?

Our response: We appreciate the reviewer’s suggestion. The exposure cohort of our study were all 20 to 45 years women who had ever pregnancy once but had abortion experience, and we collected these population from the medical record of abortion from CD-9-CM diagnosis codes 634 to 637 from the registry of NHIRD.

Point#2 Please clarify what incident diagnostic means. It includes cancer detected on in a routine screening?

Our response: We appreciate the reviewer’s suggestion. The incident cancer event, which was diagnosed from whether seeking medical advice due to physical symptoms or routine screening was identified as the first date of the female cancer diagnosis from the TCR after the index date. We made adjustment and as follows:

“The major outcome of interest was risks of incident female cancers, including breast, cervical, uterine, and ovarian cancers, comparing matched abortion and non-abortion cohorts. The incident cancer event, which was diagnosed from whether seeking medical advice due to physical symptoms or routine screening was identified as the first date of the female cancer diagnosis from the TCR after the index date.”

Point#3 Pleas clarify that if in the dataset any stage of cancer was included eg. Stage I was excluded or included in situ (HSIL) in cervical cancer. 

Our response: We appreciate the reviewer’s suggestion. The cancer diagnosed stages including carcinoma in situ, and stage I to stage IV. We made adjustment and as follows:

“The major outcome of interest was risks of incident female cancers, including breast, cervical, uterine, and ovarian cancers, comparing matched abortion and non-abortion cohorts. The incident cancer event, which was diagnosed from whether seeking medical advice due to physical symptoms or routine screening was identified as the first date of the female cancer diagnosis from the TCR after the index date and included all cancer stages such as carcinoma in situ, and stage I to stage IV.”

Point#4 In cervical cancer do you know the status of vaccination against HPV of participants?

Our response: We appreciate the reviewer’s suggestion. We had no information about the status of vaccination against HPV of participants, which could not be derived due to secondary data analysis. We made the adjustment of limitation and as follows:

“First, there were several unobservable potential confounding factors due to secondary data analysis, including family history; lifestyle factors including cigarette smoking and alcohol; vaccination status including HPV vaccination; and some hormone-related risk factors such as age at first childbirth, menarche, and menopause.”

Point#5 Table1, 2, 3 and flowchart are not included in the manuscript or as an additional material

Our response: We appreciate the reviewer’s suggestion. Our tables and figures will be present in the latter version.

Point#6 Could you explain, which mechanism could correlate abortion and breast cancer? Long exposure of breast tissue to oestrogens?

Our response: We appreciate the reviewer’s suggestion. In our study, we did find the association between abortion and breast cancer. However, it was thought the possible mechanism might be that abortion interrupted the complete differentiation of breast epithelial cells which was originally promoted by increasing estrogen and progesterone level during the full-term pregnancy and then raise the risk of carcinogenesis. We made the adjustment and as follows:

“With respect to breast cancer risk, existing studies found mixed evidence related to the associations between abortion and female breast cancers [4, 5, 19]. It was thought the possible mechanism might be that abortion interrupted the complete differentiation of breast epithelial cells which was originally promoted by increasing estrogen and progesterone level during the full-term pregnancy and then raise the risk of carcinogenesis [10, 20]. However, our current study did not find a significant association between abortion and female breast cancer, which were consistent with other previous research focusing at young women [5, 21].  ”

Reviewer 4 Report

Review of Manuscript IJERPH-2193634

Manuscript IJERPH-2193634 explores the risks of incident female cancers among women aged 20 to 45 years who underwent abortion compared with those who did not in Taiwan. The authors conducted multivariable Cox proportional hazard modeling using a longitudinal observational cohort study using three nationwide population-based databases in Taiwan, focusing on 20- to 45-year-old 27 women, with 10 years of follow-up. The authors found lower risk of uterine cancer and ovarian cancer but no significant difference in risk of breast cancer or cervical cancer among matched abortion compared with non-abortion cohorts.

This article is well-written and provides important contributions to the literature. However, three sections require minor revisions before it can be considered for publication: (1) Introduction, (2) Methods, and (3) Discussion.

In addition to this, I urge the authors to do another spelling and editing check for English-language typos and misspellings, since I found some throughout the paper.

(1)   Introduction

First, in the “Introduction” section, it would be great if the authors could state why existing studies (mentioned by the authors in lines 64-90) examining the association between abortion and various types of female cancers found inconsistent results. I think explaining potential reasons might help frame the importance of this particular study and the contributions that the authors make to this body of literature, particularly because, as the authors have stated, research in this field has been inconsistent. In addition, the authors should state the contributions of this particular piece to the literature. Drawing from the studies the authors mention in lines 64-90, what are the novel contributions of this particular study? Why is this study important and useful? How is this study different than those others cited? If previous research is limited (or focuses on something different than the current study), then I think it’s also important to state that as a novel contribution of this study to the current body of literature.

(2)   Methods

First, in the “Methods” section, please note the text in lines 113-115: “This section may be divided by subheadings. It should provide a concise and precise description of the experimental results, their interpretation, as well as the experimental conclusions that can be drawn.” Does this text need to be erased?

Second, while the authors do a very thorough job in explaining how the study population was narrowed down, I think it would be useful to explain what propensity score 1-to-3 matching entails and why it was an useful approach for this particular study. In other words, why was it important to match the abortion group with the non-abortion group? I think a further explanation of this would help the reader also sift through the results and the discussion sections.

In addition, it would be useful if they also explained what a multivariable Cox proportional hazard model is and why the authors chose this type of model to answer their research questions. Is this the best model for this type of analysis?

Finally, connected to these previous suggestions, there is limited information about how the authors employ data from the 10-year follow-up. The authors sell this study as a longitudinal study with a 10-year follow-up, but it is not clear how this follow-up plays into the analyses performed with either the propensity score 1-to-3 matching and/or the Cox proportional hazard analysis. An explanation of the role of the 10-year follow-up, as well as a discussion of the merits and contributions of this analysis would be very helpful.

(3)   Discussion

The “Discussion” needs to account for the points above, in addition to discussing major limitations and what their consequences are for the results from this study.

Overall, this paper is well-written and provides important contributions to the literature, but it requires minor revisions before it can be considered for publication.

Author Response

Point#1: First, in the “Introduction” section, it would be great if the authors could state why existing studies (mentioned by the authors in lines 64-90) examining the association between abortion and various types of female cancers found inconsistent results. I think explaining potential reasons might help frame the importance of this particular study and the contributions that the authors make to this body of literature, particularly because, as the authors have stated, research in this field has been inconsistent. In addition, the authors should state the contributions of this particular piece to the literature.

Our response: We appreciate the reviewer’s suggestion. The inconsistent results from different studies may be related to the size of eligible population, study methods, follow-up time, adjusting covariates, and particularly among different races. We revised the descriptions and as follows:

“Existing studies examining the association between abortion and other types of female cancers also found inconsistent results. For example, a prospective cohort study among 267,400 female textile workers with nearly 10-year follow-up in China found that women with abortion history was not associated with increased cancer risk but was related to significantly reduced risk of uterine corpus cancer [6]. A nationwide cohort study among 2,311,332 Danish women found that reduced endometrial cancer risk in those with pregnancy whether terminated with induced abortion or with childbirth [11]. A case-control study among women aged 50 to 74 years in Sweden found that reduced ovarian cancer risk in women with incomplete pregnancies [12], and another case-control study conducted in China found a relation between lower incident ovarian cancer and women having the history of two or more incomplete pregnancies [13]. Dick et al. (2009) conducted two case-control studies among 4,500 Australian women and found no significant association between spontaneous or induced abortions and later incident ovarian cancer for parous or for nulliparous women [14]. However, the inconsistent results regarding incident female cancers from different studies may be related to the size of eligible population, study methods, follow-up time, adjusting covariates, and even among different races.”

Point#2: Drawing from the studies the authors mention in lines 64-90, what are the novel contributions of this particular study? Why is this study important and useful? How is this study different than those others cited? If previous research is limited (or focuses on something different than the current study), then I think it’s also important to state that as a novel contribution of this study to the current body of literature.

Our response: We appreciate the reviewer’s suggestion. Although previous studies already examined the relation between abortion and female cancers, there were fewer current studies investigating related issues with longer follow-up period. In addition, lack of evidence remains regarding the association between abortion and later potential cancer incidence in Taiwan. We make the adjustment and as follows:

“Abortion in Taiwan was legalized by the Genetic Health Act of 1985. A woman can undergo abortion only under certain circumstances (e.g., medical reasons, mental health issues, or psychological impact), and must obtain the consent of her husband or the permission of her parents if she is unmarried and aged younger than 20 years [15]. Although previous studies already examined the relation between abortion and female cancers, there were fewer recent studies investigating related issues with longer follow-up period, particularly in Asian population. In addition, lack of evidence remains regarding the association between abortion and later potential cancer incidence in Taiwan. This study aimed to investigate the potential risk of abortion regarding further female cancers incidence among fertile women. Specifically, we used a longitudinal population-based cohort study with 10-year follow-up in Taiwan to compare the incident female cancers (i.e., breast, cervical, uterine, and ovarian cancers) between women who did and did not undergo abortion. ”

Point#3: First, in the “Methods” section, please note the text in lines 113-115: “This section may be divided by subheadings. It should provide a concise and precise description of the experimental results, their interpretation, as well as the experimental conclusions that can be drawn.” Does this text need to be erased?

Our response: We appreciate the reviewer’s suggestion. We will remove the formatting directions as in the ijerph-2193634 manuscript of line 111-115.

Point#4: Second, while the authors do a very thorough job in explaining how the study population was narrowed down, I think it would be useful to explain what propensity score 1-to-3 matching entails and why it was an useful approach for this particular study. In other words, why was it important to match the abortion group with the non-abortion group? I think a further explanation of this would help the reader also sift through the results and the discussion sections.

In addition, it would be useful if they also explained what a multivariable Cox proportional hazard model is and why the authors chose this type of model to answer their research questions. Is this the best model for this type of analysis?

Our response: We appreciate the reviewer’s suggestion. Propensity score 1-to-3 matching was used for generate adequate comparison groups in order to reduce the confounding bias from basic characteristics. Multivariable Cox proportional hazard model was commonly used to measure the relation between survival time and predictor variables after adjusting basic characteristics. We made adjustment and as follows:

Study population: “To compare the potential female cancer risk between comparable abortion and non-abortion cohorts, the propensity score caliper matching method with 1-to-3 match was used to generate adequate comparison groups based on propensity sore in order to reduce the confounding bias from basic characteristics. Propensity score was generated using a logistic regression model including baseline age categories, average monthly payroll groups, fertility, diabetes mellitus, polycystic ovarian syndrome, endometrial hyperplasia, endometriosis, hormone-related drugs, and Charlson comorbidity index (CCI) categories [16, 17].

Statistical analysis: “Differences in incident female cancers between the matched cohorts were analyzed using multivariable Cox proportional hazards models which was commonly used to measure the relation between survival time and predictor variables, adjusted for baseline confounding variables.

Point#5: Finally, connected to these previous suggestions, there is limited information about how the authors employ data from the 10-year follow-up. The authors sell this study as a longitudinal study with a 10-year follow-up, but it is not clear how this follow-up plays into the analyses performed with either the propensity score 1-to-3 matching and/or the Cox proportional hazard analysis. An explanation of the role of the 10-year follow-up, as well as a discussion of the merits and contributions of this analysis would be very helpful.

Our response: We appreciate the reviewer’s suggestion. As mentioned in the part of Method. We identified the abortion and non-abortion cohort from 2004 to 2007, then followed until December 31, 2017 for at least 10-year follow-up period. We conducted Cox proportional hazard to adjust the durations and the event probabilities. We made adjustment and as follows:

In the Method section:

We first identified the exposure cohort of women of child-bearing age, 20 to 45 years, who had abortion records with an induced or spontaneous abortion diagnosis based on ICD-9-CM diagnosis codes 634 to 637 in an outpatient or inpatient record from the NHIRD between January 1, 2004, and December 31, 2007 (n = 278,850). The first date of the abortion record was defined as the index date. For the non-abortion comparison cohort, we included all women aged 20 to 45 years from the NHIRD in 2004 to 2007, then excluded those who had abortion records during the entire study period until the study end date (December 31, 2017) (n = 5,001,653), in order to clearly investigate the relation between the exposure of abortion and afterward female cancer events from the comparison of 2 cohorts for at least 10-year follow-up period

In the Variable Definitions section

“To compare groups, we followed each abortion and non-abortion subject for at least 10 years from the index date to the date of incident cancer diagnosis, study end date on December 31, 2017, or death date, whichever came first. We then calculated total person-years for each study subject and cancer incidence rate per 100,000 person-years for each incident cancer event.”

Point#6: The “Discussion” needs to account for the points above, in addition to discussing major limitations and what their consequences are for the results from this study.

Our response: We appreciate the reviewer’s suggestion. We made the adjustment from Discussion and as follows:

“ The strength of the current study is that it is a first national population-base study in Taiwan examining the association between abortion and female caners. In addition, it is a prospective longitudinal cohort study using three nationwide population-based databases from 2004 to 2007 with 10-year follow-up until 2017, using propensity score matching 1-to-3 for comparison. Our comparison cohort was without any abortion record since index date to study end date to thoroughly examine the relation between abortion and afterward female cancer events in comparison for at least 10 years follow-up period. The record of abortion, deliveries, and comorbid conditions could be derived from the NHIRD, reducing the possibility of selection bias and recall bias.”

Round 2

Reviewer 2 Report

Dear authors. The MS has been updated and presentable. 

Reviewer 3 Report

The authors have addressed all my questions

Thank you so much